# Peer review of "An Early Collaborative Intervention Focusing on Parent-Infant Interaction in the Neonatal Period. A Descriptive Study of the Developmental Framework"

_ijerph, 2021, doi:10.3390/ijerph18126656_

Round 1

Reviewer 1 Report

It has been a pleasure to read the manuscript.

Congratulations to the authors for the excellent work they had done and the importance of it.

It is very well structured, perhaps I would add a diagram within the results part to make it clearer.

I think the introduction is too long. I think it could benefit to be clearer and shorter. Instead of inventing a protocol, I suggest saying build ...

The methodology is correct, but it is not well understood since everything is united, it would be appropriate to separate it into different sections: Participants, criteria, recruitment ... everything more defined.

And within the discussion part, it would include more strengths of the study. 

The results seem adequate to me, although I think they should be accompanied by some other figure so as not to be so difficult reading.
Conclusions should be more concise.

Good job!

Author Response

Thank you so much for your much valued comments on our manuscript. Our comments can be found in the file.

Reviewer 2 Report

This study aimed to describe the rationale, development, framework, and practical provision of the new early collaborative intervention program. I am glad to review this manuscript and have some comments on this manuscript.

In the Introduction section, authors have well addressed the necessity of the intervention development for moderate-to-late preterm infants in the short hospital days. However, there is lack of descriptions on theoretical backgrounds and frameworks of the intervention development. Regarding this comment, authors have described the use of the Template for Intervention Description and Replication. Readers needs to know the details of it and to identify whether it includes theoretical frameworks (e.g., attachment theory or infant-parent relationship theory).

In the Results section, 1) authors have created the name of the intervention as “Early Collaborative Intervention program”. The term of “collaborative” was indicated as the involvement of parents and preterm infants. This reviewer has noticed that the collaboration occurred among parents and medical staffs in this study. If the authors think the collaboration occurred among parents and infants, the details of the collaboration should be described. The collaboration may be defined as the integrated efforts of stakeholders to achieve a shared goal. In this context, the collaboration efforts with parents and medical staffs should be addressed in this manuscript to achieve the positive parent-infant relationship. 2) The role of each EACI-provider should be addressed in the process of parent-infant interaction during the 3 sessions. In this regard, nurses who are easily accessible to infants might have intervened the 3 sessions. Please describe about how authors confirmed the competences of each EACI-providers. 3) Authors emphasized the paternal role. Please describe in details about how the father, rather than the mother, was involved. 4) The intervention including 3 sessions should be well coordinated in the busy clinical setting. Please describe who coordinated the whole process of the intervention in the clinical setting. 5) Each EACI session should be more described systematically (this is the clinical trial-based): how nurses (or other medical staffs) are approached, what nurses (or other medical staffs) mentioned, how parents responded to infants et al. In this regard, the qualitative data are recommended in the section. 6) Please include the part of outcome measures. Authors listed up outcome measures, but this is not enough.

In the Discussion section, this study has focused a bidirectional relationship between parents and infants. In the entire section, the bidirectional relationship was not well addressed. For example, the mentions about how infants gave the signals of their needs and responded to parental behaviors should be added.

Author Response

Thank you so much for your much valued comments. Our comments can be found in the file. 

Round 2

Reviewer 2 Report

Authors have responded to reviewers' comments well. I have no more comments.